# Common Pool Resource Management: Assessing Water Resources Planning for Hydrologically Connected Surface and Groundwater Systems

**Francisco Muñoz-Arriola** [1,2,3,*] , **Tarik Abdel-Monem** [3] **and Alessandro Amaranto** [4]

1. Department of Biological Systems Engineering, University of Nebraska-Lincoln, Lincoln, NE 68583, USA
2. School of Natural Resources, University of Nebraska-Lincoln, Lincoln, NE 68583, USA
3. Public Policy Center, University of Nebraska, Lincoln, NE 68588, USA; tabdelmonem2@unl.edu
4. Department of Electronics, Information, and Bioengineering, Politecnico di Milano, Via Ponzio 34/5, 20133 Milano, Italy; alessandro.amaranto@polimi.it
* Correspondence: fmunoz@unl.edu

**Abstract:** Common pool resource (CPR) management has the potential to overcome the collective action dilemma, defined as the tendency for individual users to exploit natural resources and contribute to a tragedy of the commons. Design principles associated with effective CPR management help to ensure that arrangements work to the mutual benefit of water users. This study contributes to current research on CPR management by examining the process of implementing integrated management planning through the lens of CPR design principles. Integrated management plans facilitate the management of a complex common pool resource, ground and surface water resources having a hydrological connection. Water governance structures were evaluated through the use of participatory methods and observed records of interannual changes in rainfall, evapotranspiration, and ground water levels across the Northern High Plains. The findings, documented in statutes, field interviews and observed hydrologic variables, point to the potential for addressing large-scale collective action dilemmas, while building on the strengths of local control and participation. The feasibility of a "bottom up" system to foster groundwater resilience was evidenced by reductions in groundwater depths of 2 m in less than a decade.

**Keywords:** common pool resources; integrated water management; water governance; water resilience





## 1. Introduction

Common pool resource (CPR) institutions have been the subject of extensive research for several decades. A CPR is defined as a consumable resource where it is difficult to exclude users and where one person's use depletes the pool for others [1]. Much of this commentary has focused on what the literature calls the collective action dilemma, defined as the tendency for actors to overexploit natural resources such as water, fisheries, and grazing forage in the absence of norms and rules developed by users to govern sustainable use [1,2] and her colleagues argued that while regulation by an external authority is necessary in some circumstances, empirical evidence shows that individual users can overcome self-interest and avert a "tragedy of the commons" through collective action [3]. Based on field research in settings such as small irrigation districts, [2] identified a framework of design principles which demonstrated that users in multiple, small-scale environments have successfully created and used CPR arrangements that work to their mutual benefit.

This study adds to that research by examining integrated surface and ground water management plans (IMP) in the Upper Platte River Basin, where Nebraska employs a statutorily enacted framework for state and local government cooperation in the integrated management of surface and ground water—the Ground Water Management and Protection Act (GWMPA) (Neb. Rev. Stat. §46-701 et seq.). In examining this framework, we

relied on Ostrom's design principles for common pool resource institutions, because of its "bottom-up" perspective. Nebraska's unique system can represent an alternative to manage common pool of water resources worldwide. Water management in Nebraska includes a statewide agency, Nebraska Department of Natural Resources (NeDNR), with primary statewide authority over surface water, and 23 Natural Resource Districts (NRDs), public entities with taxing authority and primary responsibility for regulatory control over ground water (Figure 1). When state lawmakers established the NRD framework in 1972 there was a consensus that boundaries should follow surface watersheds and that local control was important to the citizens of Nebraska [4]. NeDNR and the NRDs are jointly responsible for facilitating the development of integrated water management plans.

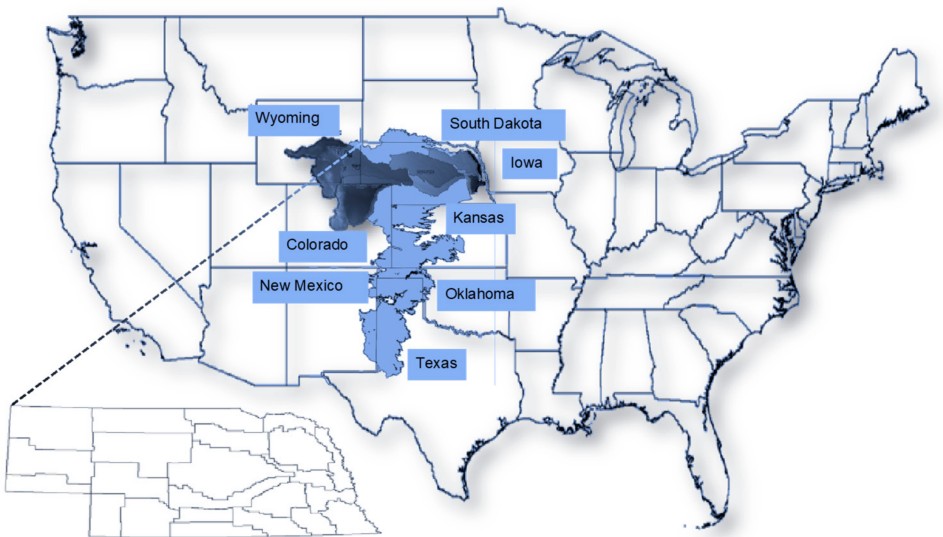

**Figure 1.** Platte River Basin (PRB) and Nebraska's natural resources. The deep blue tones evidence the main topographic features and PRB's sub basins. The light blue area is the High Plains Aquifer. At the bottom, it can be seen the state of Nebraska and its 23 Natural Resources Districts.

Examining Nebraska's approach is important for several reasons. Globally, water use for irrigation is the largest and key to develop sustainable water planning and management for food and energy production [5]. In the USA irrigation accounts for 62% of water withdrawals, being Nebraska the top state in irrigated acreage. Additionally, along with many other western states of the USA, Nebraska faces challenges in meeting competing demands for water by multiple in-state users and various interstate obligations. These challenges are exacerbated by episodes of severe drought and floods [6,7], the increasing likelihood of long-term changes in climate [8], the inherent risks to water supplies, and volatile crop markets driving resources' tradeoffs [9]. State policymakers are sensitive to the importance of managing water for its agricultural economy; however, its political culture values local control of natural resources, especially ground water, and the state has also experienced a history of conflict over water policy. These discrepancies in policies for CPR design and management water can also be evident in integrated water resources management frameworks and water governance across the globe [10–15]. CPR design principles based on principles of "bottom-up" governance are therefore a valuable lens through which to view the challenge of managing surface and ground water with a hydrological connection that can be exacerbated by a changing climate.

## 2. Building-Blocks for a Common Pool of Water Resources

*Water Resources Management and Policy in Nebraska*

The Ground Water Management Protection Act (GWMPA) was enacted in 2004 as a result of a growing recognition that Nebraska needed a strong proactive framework to

manage integrated surface and ground water. The statute was passed with widespread support in the unicameral legislature, with 44 lawmakers voting in support of the bill and only two opposed [16]. The GWMPA was the result of a consensus recommendation of a gubernatorial task force representing a diverse range of water users across the state. The task force recognized that a major issue facing the state was harm to surface water appropriations from ground water irrigation [17]. The GWMPA requires development of IMPs in areas designated as fully or over appropriated through a joint process between the NeDNR and the applicable NRD. The NeDNR designated the Upper Platte Basin (UPB) as "over-appropriated," triggering a statutory requirement for NRDs in that basin to develop individual IMPs in their jurisdictions as well as a Basin Wide Plan across the five NRDs in the UPB (Figure 2).

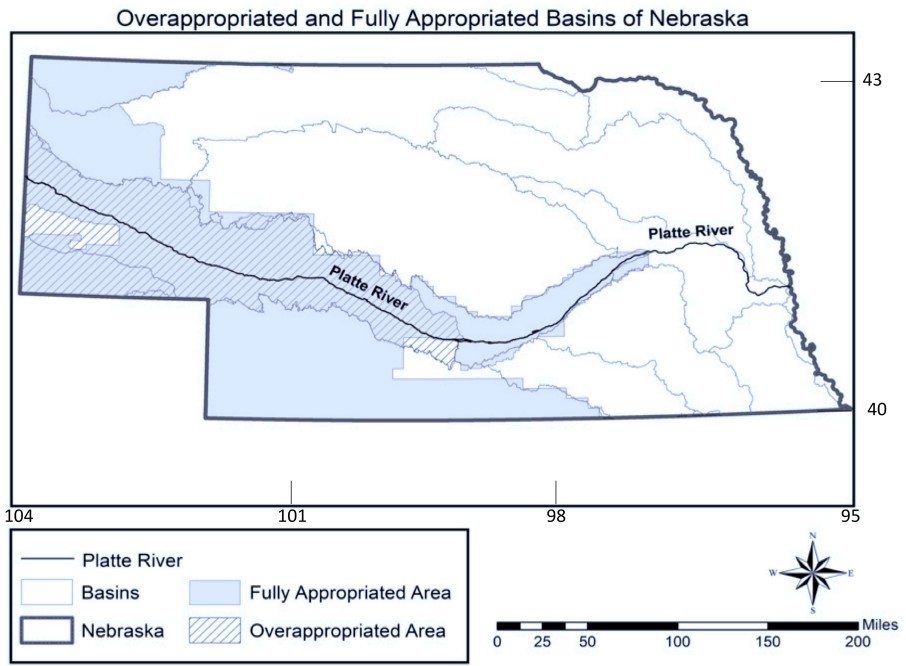

**Figure 2.** Nebraska's fully and over appropriated surface water boundaries in the Upper Platte Basin.

The IMP process creates a partnership between NRDs and NeDNR to maintain a sustainable balance between water supply and use, and to roll back over-appropriated usage to sustainable levels. Goals and objectives of the IMP are jointly determined by the NRD and NeDNR, including consultation and collaboration with stakeholders (Neb. Rev. Stat. §46-715 et seq.) Only NeDNR and NRDs have decision-making authority; however, the GWMPA requires them to consult and collaborate with public power and irrigation districts and other major stakeholders in development of an IMP. For example, the Central Nebraska Public Power and Irrigation District (CNPPID) uses surface water to generate electricity at a federally licensed hydropower dam in the UPB and delivers surface water for irrigation to over 400,000 hectares along the North Platte and Platte River sub-basins. Its service area cuts across several NRDs in the basin, and ground water use affects the delivery of surface water to irrigators served by CNPPID. The drafters of the GWMPA recognized that depletions to surface water appropriations from ground water use are a major challenge to integrated management, and that offsets to new depletions by NRDs are the primary solution to achieving a balance between water supply and use in areas with a hydrological connection [17].

NRDs are locally elected political entities whose boundaries follow the watersheds of the state's major river systems, and that develop their own priorities and programs for natural resources management based on local preferences and needs [4]. Nebraska's water governance system is unique among the fifty states. Some western states employ a highly

centralized orientation, albeit with significant consultation from local entities [18–20]. Other states—like Texas—have historically taken a much more decentralized approach, with local entities driving water use and management [21]. In Nebraska, rules for managing ground water are formally nested within a system of state-wide facilitation by NeDNR and NRDs. The approach provides for local autonomy but situates local decision-making within a vertical structure of joint decision-making with the state's NeDNR that resembles a federal system—defined by [22] as jurisdictions that are nested across levels, e.g., counties within a state.

Following the adoption of an IMP, the Nebraska GWMPA requires the NeDNR to annually evaluate the expected long-term availability of water supplies. The ultimate test of the impact of the GWMPA will be sufficiency of the water supply in the long term for beneficial uses (Neb. Rev. Statute §46-713). This test is especially important in the UPB where a significant area is over-appropriated. While the GWMPA requires the five NRDs in the UPB to develop a basin-wide plan, there are distinct differences within each district. [23] also applied Ostrom's design principles to the Platte River Basin in their study of the perspectives of water users. In the present study, the Ostrom's eight principles listed and defined below represent an opportunity to identify the interdependency between water governance and distributed ground water-surface water interactions.

1. Clearly defined boundaries: This principle states that managers should clearly define the boundary of the CPR and who has rights to withdraw resources. In the absence of clearly defined boundaries there is little incentive to coordinate, because of the risk that "free riders" will benefit from, and eventually destroy, the resource.
2. Appropriation rules relevant to local conditions: Each CPR is unique in its conditions for water use. Incentives to cooperate depend on usage rules that are reasonable and reflect the situation. A "one-size-fits all" approach to managing water supply and use discourages cooperation at the local level.
3. Participation by users: The individuals who directly interact with the CPR and with one another on a local level are in the best position to modify operations over time, and therefore they are motivated to participate in decision-making.
4. Monitoring by users: Despite shared norms valuing compliance with cooperative arrangements, most cases of long-enduring common pool resources involve active investments in monitoring by the resource users themselves. Local users are bound by these arrangements to effectively monitor the common pool resource.
5. Graduated sanctions: Punishment for non-compliance by actors in robust self-governing settings occurs in graduated steps, because local monitors are familiar with the individuals and circumstances of the infraction.
6. Accessible conflict resolution: Conflicts are often resolved informally by local leaders in robust CPR settings.
7. Recognition of local rules: External government officials recognize the authority and legitimacy of rules that are developed by local actors.
8. Nested enterprises: Established rules for management of CPRs at the local level are nested within rules at higher-level governmental jurisdictions, creating a complete system of governance.

## 3. Methodology

This study is a qualitative analysis of both the text of the GWMPA, as well as local decision-maker and stakeholder perspectives, based on an in-depth case study of one NRD in the Upper Platte River Basin. The location of the NRD and the identities of the interviewees are held confidential. The questionaries (Table 1) were part of the proposal Cross-scale Common Pool Resources Linkages in Integrated Water Management Plan reviewed by the Institutional Review Board under the IRB#745-14-EX. The authors worked independently to code provisions of the GWMPA using Ostrom's design principles as an organizing framework, and ATLAS.ti as their analysis software. In addition, there were field interviews with nine decision-makers and stakeholders to ask how implementation

was proceeding in the NRD. Interviewees were selected to represent the NeDNR, the NRD in question, and the stakeholders (i.e., users and societal sector representants) who participated in the NRD's IMP development and implementation.

**Table 1.** Sections of the Ground Water Management Protection Act (GWMPA) reflecting common pool resource (CPR) design principles.

1. Let's begin with the development of the most recent IMP. What was your overall role in the process? Have you been involved in the development and implementation of the plan? What about the role, if any, of others in your organization?
2. Did you interact with other organizations and government agencies involved in the IMP process? Who was involved from other organizations and government agencies? How often did you meet during the development of the IMP?
3. The IMP process requires a map that delineates the geographic area. Who was involved and what were the considerations that went into the map? What issues or difficulties came up in delineating the area with a hydrological connection?
4. The IMP process also requires ground water and surface water controls. Who was involved and what were the considerations that went into deciding which controls to include in the plan? What issues or difficulties came up in deciding on those controls?
5. The IMP has been in place now for at least two years. Who has been involved in monitoring water supply and use? How would you say compliance with the plan is going? Do water users think that the plan spreads the costs and benefits fairly?
6. Have conflicts between surface and water users emerged during either the development or implementation of the IMP? Have any issues arisen because of requests for new water uses that may require offsets? How are those issues resolved?
7. Let's wrap up by asking you how effective you think the IMP process has been in managing water with a hydrological connection? What has worked especially well in your view? What improvements in the process are needed in your view?

Interview questions also followed the design principles framework, in order to prompt responses about the overall operational characteristics of the IMP, such as how IMP boundaries were delineated, how monitoring and compliance mechanisms worked "on the ground," and the nature and extent of interactions between decision-makers and stakeholders. The authors also probed for interviewees' perceptions of the IMP process overall, their criticisms, and suggestions for improvements. The interview questions asked about their experiences with the full range of CPR design principles, but responses varied depending on the roles played by the interviewees. Time limitations affected the extent to which the interviews captured experiences incorporating all of the CPR design principles; the field guide allowed the interviewers some discretion on allocating time to the various questions.

Finally, changes in ground water levels and recharge were estimated as evidence of the complexity of a coupled hydrological and human system. Such monitoring integrates the potential ground water recovery in response of addressing large-scale collective action dilemmas, while building on the strengths of local control and participation in a changing environment. Ground water level changes were obtained from [24] following [7] criteria for station selection. Measurements of precipitation were obtained from the Global Land Data Assimilation System (GLDAS [25]). The consumptive use of water was estimated from the MODerate resolution Imaging Spectroradiometer (MODIS [26]). Recharge was determined as the difference between precipitation and evapotranspiration assuming a constrained runoff generation at the location of the well. The recharge is normalized using statistics of dispersion (standard deviation) and central tendency (mean) obtained from data spanning between 2002 and 2010 (due to the availability of MODIS-ET).

## 4. Results and Discussion

The Legislative Framework is integrated in Table 2 indicating each of Ostrom's principles, and corresponding sections of the GWMPA, including requirements for IMPs. The most salient individual sections of the statute are identified for each principle. It should be noted that the last of Ostrom's principles—recognition of local rules and nested enterprises—

are not coded because their overall design purposes are integrated through the overall approach requiring state (NeDNR) and local (NRD) coordination and cooperation in IMP operationalization. This cross-jurisdictional approach recognizing both local and state responsibility is clearly reflected in the statute's legislative findings, which identify NRDs as the "preferred regulators" for groundwater (§46-702), and state that the objective is that "(a)ll involved natural resource districts, the department, and surface water project sponsors should cooperate and collaborate on the identification and implementation of management solutions" (§46-703 (6)).

**Table 2.** Sections of the GWMPA reflecting CPR design principles.

| **Boundaries** | §46-715(1)(a), §46-715(1)(b), §46-715(2)(b), §46-718(2) | IMPs are mandated in over appropriated or fully appropriated areas as agreed-upon by NeDNR and impacted NRDs. |
|---|---|---|
| **Appropriations** | §46-715(2)(c), §46-715(2)(d), §46-715(4), §46-715(5)(c), §46-716(1)(b), §46-716(1)(c), §46 716(1)(d), §46-716(2), §46-718(2), §46-739 | IMP must include one or more controls on both surface and ground water appropriation or use to sustain a balance between hydrologically connected water uses and supplies so that the economic viability, social and environmental health, safety, and welfare of the basin be achieved and maintained. Further, IMPs in over-appropriated basins must identify the amount of water necessary to offset the impact of stream flow depletions initiated after 1997 [1]. |
| **Participation** | §46-715(3)(f), §46-715(5)(b), §46-715(5)(d)(ii), §46-717(2), §46-719(3), §46-719(4) | Stakeholder groups must be consulted with during development of the IMP. NeDNR and the NRDs may amend an IMP at annual review, for which there are no provisions for involving stakeholder groups. |
| **Monitoring** | §46-715(2)(e), §46-715(3)(d), §46-715(5)(d)(ii), §46-715(5)(d)(iii), §46-715(5)(d)(v), §46-715(6) | NeDNR and NRDs jointly progress toward meeting IMP goals and objectives. NeDNR forecasts the maximum water volume from stream flow for beneficial use in both the short and long term. |
| **Sanctions** | §46-707(1–3), §46-708(3), §46-745(1), §46-745(2)(a), §46-746 (1–2) | NRDs may require reporting, metering or decommission of wells, issue cease and desist orders, initiate lawsuits, and take other forms of action. |
| **Conflict Resolution** | §46-715(5)(b), §46-718(3), §46-719(2), §46-719(3), §46-719(4) | If the parties reach agreement on the plan, then the NeDNR and the NRD adopt it. NeDNR and NRDs develop and adopt the plan if participating parties disagree. If NeDNR and NRDs are in dispute, the matter may be taken to the Interrelated Water Review Board. |

[1] The year 1997 refers to the signing date of the Cooperative Agreement that created the Platte River Recovery Implementation Program (PRRIP) beginning on 1 January 2007. The PRRIP covers the Basin of the Platte River within Colorado, Wyoming and Nebraska. Each state was responsible for developing a plan to mitigate effects of surface and ground water depletions initiated after 1997.

Six Ostrom design principles include references and field interviews relevant to sections of the GWMPA. The last two principles, Recognition of Local Rules and Nested Enterprises did not receive any mention from the interviewees.

### 4.1. Clearly Defined Boundaries

The GWMPA requires IMPs to include designation of the geographic area and inclusion of a map delineating its boundaries (Neb. Rev. Stat. §46715(1–2), §46-718(2)). The IMP includes a map of the geographic area covered and delineated over-appropriated and fully appropriated portions identified through modeling efforts, each of which is subject to different requirements. The boundary and associated regulations limit water use to those who have agreed to self-regulate ground water irrigation.

Almost all interviewees indicated that establishing a geographic basis for regulatory action was a key step to the IMP. Throughout plan development, participating decision makers and stakeholders were involved in modeling efforts to measure and identify areas under their jurisdiction that were hydrologically connected, and the extent to which those areas were fully or over appropriated. These modeling efforts are ongoing and have resulted in analysis of hydrological and geological conditions in the entire basin that are incorporating groundwater flow, soil-water balance, and surface water dynamics. These modeling efforts have been supported by multiple sponsors, including NRDs, state agencies, municipalities and power companies in the Platte River Basin, and have resulted in identified geographic boundaries of the IMP and extensive data on its hydrological characteristics that have driven decision making.

**Mandatory IMP Interviewee #6.** "The COHYST [Cooperative Hydrology Study] group, which stands for the conjunctive cooperative hydrology study group, which involved game and parks, DNR, all the NRDs, the two major irrigation districts, CNPPD and NPPD, kind of make up the COHYST study stuff. The Platte River program headwaters group is somewhat involved as well. We were developing the tools and DNR basically requested that we do the study, the COHYST group. So, we took the groundwater models to COHYST, and they ran all the models to generate the percent depletion by use".

**Mandatory IMP Interviewee #4.** "(T)he NRD didn't really have much control over the surface water. But then once they established the relationship in the COHYST between how groundwater pumping depletes the surface water. They became much more involved".

**Mandatory IMP Interviewee #5.** "Every 40-acre tract out here has a designated value that they have worked out through this COHYST model that shows the returns and the length of time that . . . obviously closer to the river water would get back there faster obviously than it would next to the canal...".

*4.2. Appropriation Rules Relevant to Local Conditions*

The GWMPA mandates that IMPs include one or more controls on surface and ground water appropriation or use to sustain a balance between hydrologically connected water uses and supplies, and to maintain the economic viability, social and environmental health, safety, and welfare of the basin. Further, IMPs in over-appropriated basins must identify the amount of water necessary to offset the impact of stream flow depletions initiated after 1997 (see §46-715(1–6)). The year 1997 refers to the signing date of the Cooperative Agreement creating the Platte River Recovery Implementation Program (PRRIP) beginning on 1 January 2007. The PRRIP covers the Platte Basin within Colorado, Wyoming and Nebraska. Each state is responsible for developing a plan to mitigate effects of surface and ground water depletions initiated after 1997. Thus, the PRRIP and IMPs in the Upper Platte Basin are interconnected documents.

The NRD's fully appropriated portion is under a moratorium on new well permits and expanded irrigation acres as per statutory requirements (§46-714(1–2)). The NRD is responsible for offsetting new or expanded ground water irrigation, as well as increases in consumptive municipal use from population growth and commercial/industrial consumptive use, up to limits of 25 million gallons per year. The NRD is also responsible for finding offsets to new or increased non-municipal industrial use up to 25 million gallons per year. The NeDNR has also placed a moratorium on new surface water appropriations. The over-appropriated portion is under the same moratorium; however, the NRD must also offset "new" depletions dating back to 1997. Appropriation rules allow for continued development through the use of offsets to new or expanded uses. One example of strategies to offset new depletions in the over-appropriated area is an agreement between the NRD and local irrigation districts. Surface water irrigators may switch to their (existing) wells, and the NRD applies to the NeDNR on their behalf for the right to divert excess river flows into canals for ground water recharge and retiming base flows to the river. NeDNR calculates the addition to the base flow and counts it as an offset to new depletions.

Participants had mixed but generally positive perceptions about appropriation rules and their relevance to local conditions. The IMP mandate to decrease over-appropriation drives restrictions and controls in the area, but also allows for collaborative mechanisms among decision makers and stakeholders to establish use arrangements that comply with IMP goals. This has led to the creation of some cooperative projects between the NRD and stakeholders that were perceived as win-win efforts to advance both the interests of water users in the basin, as well as overall IMP goals.

**Mandatory IMP Interviewee #6.** "Basically, we have an agreement with each of the irrigation districts . . . . We have a lease agreement to put together the water rights, transfer the water rights. The irrigation district signs them, and we send them in. They total up the bills (for canal repairs) and we go half and half. They pay half and we pay half".

These agreements emerged based on trust after years of discussions: surveys of the land area; and calculations based on a hydrologic model of the interactive effects of surface and ground water in that area. Overcoming distrust between surface and ground water users took time, as did negotiations based on an equitable sharing of the investment costs associated with maintaining the canals for recharge purposes, and future benefits of the revenues from leasing unused surface irrigation water for other uses. While the agreements between the NRD and local irrigation districts require NeDNR approval to transfer surface water rights, and involve a lengthy approval process, the IMP facilitates implementation because it allows the DNR to treat transfers as a beneficial use. NeDNR's role is therefore one of facilitating the strategies developed at the local level by the NRD and irrigation districts. Thus, while the threat of regulatory controls on ground water irrigation may have been a prime motivator in bringing people together in the IMP process, local cooperation resulted in a proactive approach to controls on appropriations that were unique to local conditions and which mitigated conflict with some, though not all, users.

**Mandatory IMP Interviewee #1.** "I think the nice thing about what they are doing is that they have become partners with the surface water folks, who at the beginning of this process, when we started IMP, they were still not partners. They were still thinking everyone was out to get them".

*4.3. Participation by Users*

The GWMPA mandates that stakeholder groups be consulted during development of the IMP (see, e.g., §46-715(3)(f), (5)(b) and §46-717(2)). The IMP reflects statutorily mandated decision-making by NeDNR and the NRD; requires meetings with stakeholders; and outlines the process for NeDNR and NRD to annually review the progress of the IMP and jointly agree upon any amendments. Although the NeDNR and the NRDs may amend an IMP at annual review, there are no explicit provisions for involving stakeholder groups in the amendment process (see §46-715(5)(d)(ii)). The goals and objectives for this IMP, as well as the major strategies for addressing depletions to the Platte River, evolved from ideas discussed among NRD staff, irrigation district board members, and municipal officials prior to the start of the planning process. During the planning process, the NRD held public meetings for stakeholders and members of the public. These meetings fulfilled the consultation requirements in the GWMPA.

Decision maker and stakeholder perspectives on participation varied widely. One interviewee reported that his engagement with the NRD and other stakeholders predated the IMP, and that a great deal of mutual exchange and education among surface and ground water users had already occurred. On the other hand, another interviewee reported that those who proposed increasing minimum accretions to stream flow were "laughed off the floor." Still another questioned whether the NeDNR and the NRD actually consulted and collaborated with stakeholders to a meaningful degree, as opposed to simply gathering input and then writing the plan on their own.

The most frequent comment was that IMP stakeholder meetings were infrequent compared to the other IMPs in the western part of the state, and that consultation was perfunctory. Several reported attending and listening, without offering any input. Some stakeholders had specific ideas to propose but had the sense that the NRD was controlling the agenda. These perspectives seem to reflect characteristics of the GWMPA that restrict decision making to select entities, or do not adequately define what appropriate collaboration is among stakeholders in the IMP:

**Mandatory IMP Interviewee #8.** "Collaboration in this sense was basically, "We will meet with you and take your input." We were told many times during the (name of NRD redacted) IMP process that the NRD board would make the decisions. We sent in comments. My recollection was that the NRD drafted the IMP and presented it to the stakeholders. In many cases the department responded the same as the stakeholders did. Everyone was feeling their way. There was no set process".

**Mandatory IMP Interviewee #7.** "The statutes say that they are to consult and collaborate with us. Those are two different words. They have two different meanings. And very often what we find is, they come and consult, and they say, "We are consulting and collaborating with you now." And we would often ask, "Where is the collaboration? Where is the part where you are asking us to be involved with and participate in finding solutions to this? Because it seems like really what you are doing is consulting only".

Other interviewees who participated in various IMP development meetings believed that the highly technical nature of discussions impeded participation. As one stakeholder commented:

**Mandatory IMP Interviewee #9.** "And I do know that the water professionals and irrigators came. I think the process would have benefited from a much more educational bent. Because not everyone was on the same level of education on how water works and how this whole thing gets put together. There was very little if I remember it right, very little effort to bring people up to speed with all the stakeholders in fact. And I think I came at it with a fairly decent knowledge, but there was a lot of jargon and acronyms and things like that that probably limited how well people could participate".

*4.4. Monitoring by Users*

The GWMPA mandates that NeDNR and NRDs jointly progress toward meeting IMP goals and objectives (§46-715(3)). NeDNR forecasts the maximum water volume from stream flow for beneficial use in both the short and long term. In the IMP, the NRD tracks yearly certification of ground water use, water well construction, and consumptive uses by municipal and non-municipal industrial water systems within its jurisdiction. It also tracks the number and location of retired irrigated acres and offsets for new uses, including depletions dating to 1997 in the over-appropriated part of the basin within its jurisdiction (§46-715(2)(e)). NeDNR tracks changes in permits for surface water (§46-716). The NRD board is elected by local ground water users.

**Mandatory IMP Interviewee #1.** "So, there is a reporting and monitoring section in the plan. So basically, the NRD and my department come together and say, "OK here are all the activities that have taken place in the last year" just in a checklist fashion, have we caused more depletions? Are there more accretions? Where are we in the permitting process? And that is telling us on an annual basis are we getting where we want to be".

The NeDNR relies on NRD records for tracking certified acres, including transfers from a water rights holder associated with retired acres and/or transferred ground water use from one tract of land to another. The NRD also uses aerial photography to insure that irrigators are staying within their certified number of acres. At the time of this study, decision-makers were finalizing plans to run an updated ground water (hydrologic) model in order to verify the number of acre feet per year that will be needed to offset depletions dating back to 1997. However, some interviewees expressed skepticism of benchmarks and incremental approaches used for monitoring and assessment of accretions or depletions under the IMP, believing that the GWMPA should require NRDs to offset depletions dating back prior to 1997, because there were prior (surface water) appropriations predating the introduction of widespread use of central pivot irrigation that were impacted by those

ground water wells. The GWMPA, however, requires only voluntary efforts to offset depletions prior to 1997 as part of an incremental approach. (§46-715(5)(d)(i)).

> **Mandatory IMP Interviewee #2.** "What is fully appropriated? Is it where your development is affecting streamflow? These are measures of degree. In our mind, what is that difference? The fact that it wasn't (fully defined) when all these plans were done was disappointing, and of major concern to us. The difference between fully and over. We still don't agree with the way the department is proposing to do that. Basically, we are not really allowed to participate any more".

Those concerns were connected to perceptions that the structure of the IMP did not allow for full participation among all stakeholders, especially surface water providers, and that there were few avenues to air such grievances. Thus, perceptions of the efficacy of monitoring activities varied depending on the interests of those involved and whether those interests were represented in the development and implementation of the IMP.

### 4.5. Graduated Sanctions

The GWMPA identifies a variety of sanctions on individual water users for non-compliance with its mandates (see, e.g., Neb. Rev. Statute §46-746). The NRD was in the early stages of implementing its IMP during this study, including monitoring for compliance with ground water controls. Nevertheless, the NRD adopts the IMP, including the moratorium on ground water use, in consultation with local users and stakeholders. Thus, local monitors are familiar with the individuals and circumstances of the infraction. In fact, several interviewees pointed to the IMP and the importance of enforcement as a hedge against further restrictions on ground water use.

Interviewees had little more to say about sanctions, partially because the ten-year timeframe to review progress towards plan goals had yet to occur. As the first increment of the current IMP was due at the end in 2019, it is possible that developments in regard to non-compliance and sanctions may emerge after the formal technical analysis of plan progress is completed. As discussed previously, there were concerns that the statutory framework of the GWMPA failed to address the effects of ground water depletions on prior (surface water) appropriations that predated the 1997 benchmark in the law.

> **Mandatory IMP Interviewee #1.** "The way the law is set up, this first increment, which is 10 years, is that we will get back to 97. So the triggers you see are built to get back to 1997. But there are still shortages in the system just because we are still ... well you have drought anyway, and you have wells that have existed before 1997 that are impacting the system as well, and those are not at this point being addressed. The interests in the part of the surface water parties is that those should all be addressed right away. But the plan isn't set up that way, its set up to do it in incremental fashion, so there is conflict and tension going on there. So it's not that there isn't a shortage, it's that we don't have to address all the shortage right now".

### 4.6. Accessible Conflict Resolution

The GWMPA provides for conflict resolution as a two-step process. If the parties reach agreement on the plan, then the NeDNR and the NRD adopt it (§46-715 (1)(a)). If NeDNR and NRDs are in dispute, the matter may be taken to the Interrelated Water Review Board (§46-719(2)). Interviewees expressed a range of opinions about the accessibility of conflict resolution mechanisms through the current plan structure. Not surprisingly, perceptions of conflict resolution mechanisms varied depending on an interviewee's overall perceptions of how well the plan advanced either individual interests, or the overall goals of the IMP to reduce over-appropriation. For example, there was general agreement that the conjunctive management approach of the IMP was beneficial, and that individual actions taken under the plan were successful.

**Mandatory IMP Interviewee #1.** "I think it's going very well. It's nice to see everyone being very conscientious about what the plan says, and how to be in compliance with that plan. (Name of NRD redacted) has been making great strides to get all those conjunctive management pieces in place. When they started the process, they purchased a lot of easements and buying out groundwater and surface water rights and retiring them. So, they have been a leader in Platte NRDs in implementing various types of practices in getting us to where we need to be".

**Mandatory IMP Interviewee #7.** "Often, we are in disagreement in terms of whether they have actually set something that will actually meet their objective. But it sounds like their objective, their intent, in areas that are not yet fully appropriated . . . try and identify where they will occur, try and head them off. That's good. But that's not really our area. Our concern is they are not really directly trying to resolve the conflict that was already created. We think that there is an obligation to try to do that".

However, there were distinct criticisms about the scope of the plan's mandate as well as a perceived absence of a mechanism to resolve issues before the ten-year increment ends. Interviewees who represented surface water users indicated that although current conjunctive management practices under the IMP were generally positive, the GWMPA does not adequately address perceived inequities between ground and surface water purveyors that existed prior to the enactment of the 2004 GWMPA amendments, because the law calls only for voluntary efforts, subject to the availability of funds, to offset depletions to streamflow dating back prior to 1997 (§46-715(5)(d)(i). Another criticism was that the plan's ten-year incremental structure does not provide an adequate means to resolve conflict that would happen in the interim period before the first ten-year phase would end.

**Mandatory IMP Interviewee #2.** "What is the best way to achieve results then? Is it to get things out there, or just wait 10 years until they get new plans done and then hope we potentially see some change? Then you see lower lake levels. Do you just have to say, 'I will keep quiet and wait my ten years because that is the only option that is out there?' We are disappointed in those options".

Both stakeholders and decision makers voiced concern that conflict management under the current GWMPA statute—and IMPs derived from its requirements—were not sufficient. For example, the Interrelated Water Review Board has never been convened, nor does it review disputes that a stakeholder may have in regard to the Plan. On the contrary, the statute and Plan allow for the NeDNR and NRD to move forward with the plan regardless of whether all stakeholders impacted by it are supportive:

**Mandatory IMP Interviewee #1.** "Well there is the Interrelated Water Review Board. Yeah, that is not if the stakeholders can't agree, but if the department and the NRD can't agree what the plan should be. As we go through the stakeholder process even with the consultation and collaboration the statutes clearly say that DNR and NRD can go back and say, 'OK you guys couldn't agree, so we are going to see if we are going to agree,' and so that is where we ended up. We could agree, so we could move forward with that plan even though not all the stakeholders were on board with it".

### 4.7. Recognition of Local Rules

The GWMPA establishes a joint decision-making process between state and local officials, and the NeDNR facilitates the planning process and approves local IMP plans.

### 4.8. Nested Enterprises

Rules for developing and implementing IMPs are nested within provisions of the GWMPA. [2] saw the volatility of climate as a common characteristic across multiple CPRs. However, [27] reflect on the absence of environmental accounting within Ostrom's CPR

design principles. In our case, the changes in ground water levels and recharge in three NRDs encompass the complexities of interdependent hydroclimate, water management, and soil physical properties [6,7,28] used crops' evapotranspirative demands, rainfall, and streamflow as variables that integrate irrigation management, rainfall variability, and soil infiltration capacities. These observed data were inputs to a data-driven model developed that successfully reproduces the changes in groundwater well-levels. Thus, the difference among Figure 3A–C can represent the differences in water governance across UPB's NRDs. For example, consistent depletion of groundwater levels in NRD-1, -2, and -3 (2, 4 and 5 m, respectively) started in 2002 indicate how consistent streamflow withdrawals affect aquifer recharge. On the other hand, inflections in the negative trends that occurred in 2005 and 2007 in the NRD-1 and the NRD-2, respectively, illustrate how changes in diverted excess of river flows increase aquifer recharge. In comparison, the rise in ground water well levels in the NRD-1 (2 m) responds to the agreement among the NRDs, irrigation districts, and NeDNR. In locations like the NRD-2 and NRD-3, the less conspicuous rise (<1 m) may be attributed to intraseasonal increments in rainfall. Depletion of ground water depth after 2012 can be attributed to droughts. The 2012 flash drought reported by [29] was evident in NRD-1, -2, and -3.

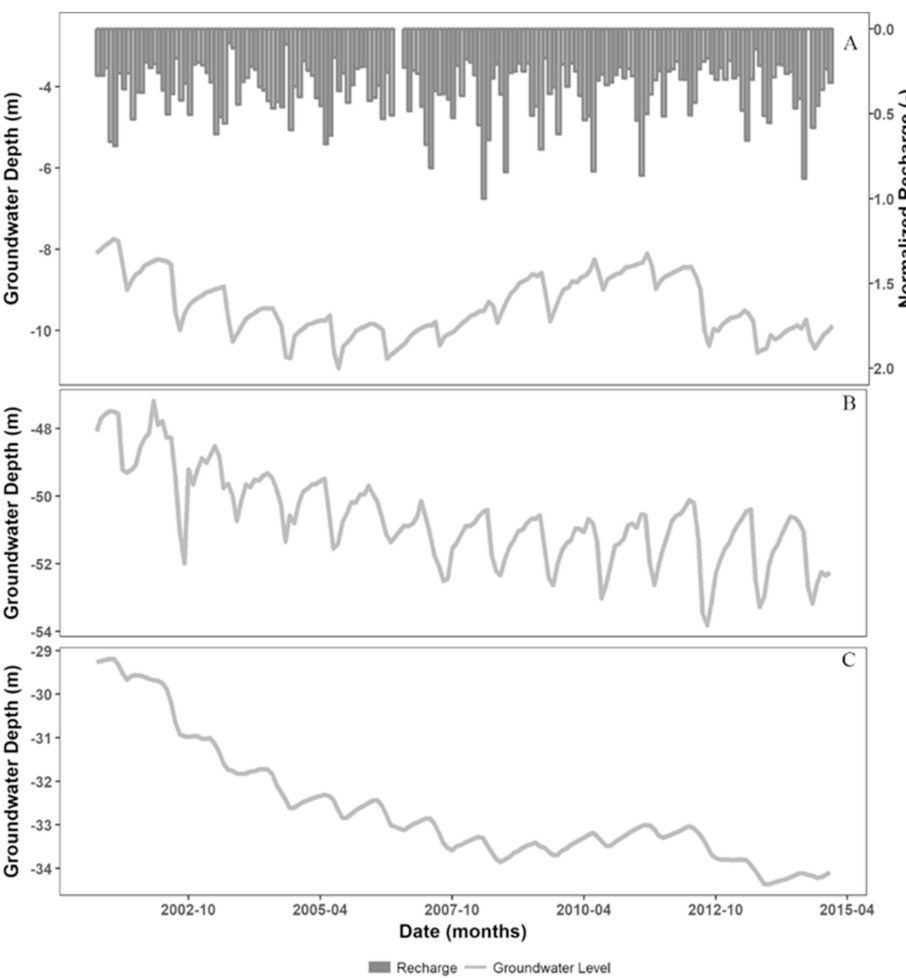

**Figure 3.** Interannual changes of integrated (surface and sub-surface) hydrological responses in three Natural Resources Districts. (**A**) illustrates NRD 1; (**B**) illustrate temporal changes in groundwater well levels in NRD 2; and (**C**) illustrate temporal changes in groundwater well levels in NRD 3. The location of the NRD is not disclosed due to security constrains.

The GWMPA, its IMP process, and data on variations in hydrological connectivity across the UPB provided the background for this study. Nebraska's decentralized frame-

work in which each NRD develops an IMP based on its unique conditions, is consistent with Ostrom's "bottom-up" approach to the management of common pool resource institutions, although this study makes no claim that following Ostrom's design principles is a predictor of successful outcomes. As [30] point out, they are relevant for simple common pool resources, but additional research is needed in more complex social-ecological systems. Conditions in the UPB are more complex than the small-scale irrigation districts that were the focus of Ostrom's original work. Nevertheless, these design principles can work as a heuristic device—helping to focus on key elements in common pool resource management like surface and ground water having a hydrological connection.

Common pool resource principles help to explain why it is in the collective interest of actors to decrease the likelihood of exploiting and exhausting resources, thereby obtaining long-term benefits for all [2]. The emphasis in Ostrom's original work was on processes of local self-governance in small-scale situations. As [31] point out, however, Ostrom recognized that when local common pool resources are part of larger systems, the organizations that govern them are more successful when linked in a nested fashion, that is, when actors at different scales share rules or strategies through formal means. Following [31], we argue that Ostrom's design principles can be used to examine the IMP process in Nebraska.

## 5. Conclusions

Water governance structures like those in Nebraska indicate that, even in a large-scale, complex common pool resource such as hydrologically connected surface and ground water, a "bottom up" system is feasible. Evidence from the field interviews suggests that local ground water users have accepted the moratorium imposed by the NRD, because it reduces uncertainties about the future, including the possibility of more devastating restrictions if previous patterns of consumptive use had continued unabated as those observed in Figure 3B,C. The exception to these findings, however, is that the GWMPA framework limits participation by surface water providers in the IMP process. This limitation becomes relevant in cases where surface water contributes to the recovery of ground water levels (Figure 3A).

Interviews revealed concern among some stakeholders with this arrangement, because they perceived it as resulting in a less-than-equitable process and outcome in terms of water appropriation. This tension between surface and ground water providers and users stems from the bifurcated system of water laws in Nebraska. Laws governing surface water use according to the doctrine of prior appropriation with its associated principle of seniority evolved independently of doctrines of reasonable use and correlative rights governing access to ground water in the state. An over-appropriated designation requires offsets to depletions of surface water flows from ground water use dating back only to 1997, even though there are older surface water appropriations impacted by those earlier depletions. Conflicts stemming from this bifurcated system, especially in over-appropriated areas, are beyond the scope of the GWMPA and the IMP process [32].

The tension that results from this bifurcated system has complicated the implementation of the IMP process. In fact, the significance of the hydrological connection between surface and ground water wasn't fully appreciated by decision-makers during passage of the GWMPA and the framework splitting jurisdiction between NeDNR and the NRDs [4]. As a result, there is a gap in the alignment of the legal framework with CPR design principles, in particular the principle that individuals who directly interact with the common pool resource and with one another are in the best position to modify operations over time, and that they are therefore motivated to participate in decision-making [2]. As the experiences of some interviewees have suggested, their participation in the development and implementation of the IMP is limited in scope, especially decisions about the extent of controls on ground water use that impact surface water supplies. These limitations, in turn, affect perceptions of an inequitable system for imposing sanctions and resolving conflicts. Ultimately, these limitations could impact the sustainable management of hydrologically connected surface and ground water supplies in the UPB.

The effectiveness of integrated management strategies may depend on the extent of connectivity and the inherent complexity of the drivers of ground water-level changes. Nonetheless such analysis is beyond the scope of this study it is evident that approaches such as those proposed by [6,7,28] could be predict changes in ground water levels in response to water policies and climate variability, and consequently the CPR design principles.

Ostrom's principles emerged from years of empirical research demonstrating their effectiveness as an alternative to hierarchical government or private market allocation of common pool resources. This study used Ostrom's framework to study the implementation of the IMP process in the field, and it identified major areas of alignment suggesting that there is potential for Nebraska's decentralized approach to achieve sustainable levels of surface and ground water. Nevertheless, the future effects of severe droughts and long-term climate change are largely unknown at this time, and more comprehensive reforms may be necessary to involve surface water providers and users in the integrated management of hydrologically connected common pool water resources. These efforts should lead to the design and creation of a more climate-resilient water infrastructure based on a better understanding of the socio-ecological functionalities of the surface water and ground water resources.

**Author Contributions:** Conceptualization, F.M.-A. and T.A.-M.; methodology, F.M.-A., T.A.-M. and A.A.; software, F.M.-A., T.A.-M. and A.A.; validation, F.M.-A., T.A.-M. and A.A.; formal analysis, F.M.-A., T.A.-M. and A.A.; investigation, F.M.-A., T.A.-M. and A.A.; resources, F.M.-A., T.A.-M. and A.A.; data curation, F.M.-A., T.A.-M. and A.A.; writing—F.M.-A.; writing—review and editing, F.M.-A. and T.A.-M.; visualization, A.A. and F.M.-A.; supervision, F.M.-A. and T.A.-M.; project administration, F.M.-A. and T.A.-M.; funding acquisition, F.M.-A. and T.A.-M. All authors have read and agreed to the published version of the manuscript.

**Funding:** The authors wish to thank the Nebraska Department of Natural Resources for funding this research. Some research ideas and components were also developed within the framework of the USDA National Institute of Food and Agriculture, Hatch project NEB-21-166 Accession No.1009760, the Robert B. Daugherty Water for Food Global Institute at the University of Nebraska and the University of Nebraska-Lincoln Institute of Agriculture and Natural Resources-Agricultural Research Division.

**Institutional Review Board Statement:** The study was conducted according to the guidelines of the Declaration of Helsinki, and approved by the Institutional Review Board (or Ethics Committee) of University of Nebraska-Omaha Institutional Review Board (IRB#745-14-EX).

**Informed Consent Statement:** Informed consent was obtained from all subjects involved in the study.

**Data Availability Statement:** Restrictions apply to the availability of interview data. As reflected in our Institutional Review Board protocol, the interview data was obtained from individuals on a confidential basis, and further disclosure of this data could directly or indirectly reveal their identities.

**Acknowledgments:** The authors also wish to thank Jesse Bradley and Jennifer Schellpeper from NeDNR for fact-checking the original report.

**Conflicts of Interest:** The authors declare no conflict of interest.

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
