# Peer review of "Common Pool Resource Management: Assessing Water Resources Planning for Hydrologically Connected Surface and Groundwater Systems"

_hydrology, doi:10.3390/hydrology8010051_

Round 1
Reviewer 1 Report
Provide a graphical abstract for better understanding.
Result section required some figures and tables, please provide your result as a schematic form in some figures.
Materials and Methods should be extended, please use references, formula and also more details about methods and data.
Use some newly literature review.
This manuscript is a bit far from standard form, and at least it needs 2 (or 3) round revisions. Then for now please handle mention above comments. More details in next round will be informed (according to your revised file).
Author Response
Dear Reviewers and Editors,
We appreciate the valuable comments you made by. The sections were reviewed and reorganized based on their comments, leading to a more robust and fluent narrative. All the changes made were tracked and observed in the version “REED et al Hydrology CPR_HYDROLOGY-REPLAY-REV-TRACK.” Below, you can see our answers to each of your comments.
Best regards
Francisco Muñoz-Arriola, PhD
REVIEWER 1
Comments and Suggestions for Authors
The work is interesting and covers Platte River Basin (PRB) and Nebraska's Natural Resources. However, the results of the work can be extended to other parts of the world. Despite this, the work has many problems that I have highlighted in the text of the paper
We provided along with the narrative elements that refer to the potential to transfer the work presented here to other locations worldwide (Line 66). In particular, the policies of surface and ground water management and the use of publicly available data (i.e., well-level, evapotranspiration, and rainfall) (Lines 278-290). We appreciate the comments made through the text. We addressed all of them but one "Ostrom (1990) and her colleagues…" because we consider that such citation represents a compilation of multiple works that Ostrom herself acknowledged in the book (Line 32). If you think the comment stands, we will be happy to change the text.
The narrative has few changes in the sections to facilitate the flow and reflect the manuscript's content. Please, see the tacked-changes version where differences are highlighted, and few sentences added. We consider that the present version of the text communicates better our findings and analyses.
THANK YOU!

Reviewer 2 Report
This manuscript investigates the topic of common pool resource management, providing and examination of integrated surface and ground water management plans in the Upper Platte River Basin. Proposed topic is very relevant and coupled use of proposed approach with applied methodology can led to useful applications in environmental and institutional fields. The paper is technically sound, and certainly of interest for readers.
However, to my opinion a certain number of questions need to be addressed, in order to improve the overall quality of the manuscript.
Broad comments:
- Abstract: goals, approach and main findings of proposed research should be better highlighted;
- Introduction: to me, it should be reformulated, clearly explaining motivations of this study and its role in scientific literature, concluding the paragraph with a summary of the paper. Furthermore, sub-paragraphs 1.1 and 1.2 could be moved into another paragraph. For example, par. 1.1 could be integrated into a single “case study” paragraph, while par 1.2, which is a list of Ostrom’s eight principles, can be reported in par. 2 or as a table.
- Results and discussion: number of questions and size of this paragraph suggest that a separate discussion paragraph should be better for providing to the reader a comprehensive vision on your work, as you do in the Conclusion section.
- In the Conclusion section I would like to be highlighted in details role, contribution and possible implications of this research.
- Acronyms: this paper reports a number of acronyms, often and repeatedly cited into the text, strongly affecting its readability. I suggest to add a table with all acronyms and a diagram that shows links between institutions and laws.
Specific comments:
- Figure 1: could you use more evident colors?
- Figure 2: please use different colors, and specify on the map the geographical position of the basin
- Figure 3: it is not clear where the contribution of "Integrated hydrological processes" emerges within the figure. It seems that it should be with "normalizd recharge", that have no definition or links in the text
- Lines 132-135: could you provide a reference for this statement?
- Lines 139-141: I think that it should be moved in the Conclusion section
Author Response
Dear Reviewers and Editors,
We appreciate the valuable comments you made by. The sections were reviewed and reorganized based on their comments, leading to a more robust and fluent narrative. All the changes made were tracked and observed in the version “REED et al Hydrology CPR_HYDROLOGY-REPLAY-REV-TRACK.” Below, you can see our answers to each of your comments.
Best regards
Francisco Muñoz-Arriola, PhD
REVIEWER 2
This manuscript investigates the topic of common pool resource management, providing and examination of integrated surface and ground water management plans in the Upper Platte River Basin. Proposed topic is very relevant and coupled use of proposed approach with applied methodology can led to useful applications in environmental and institutional fields. The paper is technically sound, and certainly of interest for readers.
However, to my opinion a certain number of questions need to be addressed, in order to improve the overall quality of the manuscript.
Broad comments:
- Abstract: goals, approach and main findings of proposed research should be better highlighted;
The abstract has two additional sentences to address this comment (Line 19-21 and Lines 23-25).
- Introduction: to me, it should be reformulated, clearly explaining motivations of this study and its role in scientific literature, concluding the paragraph with a summary of the paper. Furthermore, sub-paragraphs 1.1 and 1.2 could be moved into another paragraph. For example, par. 1.1 could be integrated into a single “case study” paragraph, while par 1.2, which is a list of Ostrom’s eight principles, can be reported in par. 2 or as a table.
Please, review the changes made to the paper. We move portions of the Introduction (sections 1.1 and 1.2) to the Results and Discussion since the information emerged from the proposed approaches.
We enhanced the Methodology section, specifically in estimating the temporal changes of Normalized Recharge and Ground Water levels. Also, we added what now is Table 1, which includes the questionaries and the corresponding IRB#.
- Results and discussion: number of questions and size of this paragraph suggest that a separate discussion paragraph should be better for providing to the reader a comprehensive vision on your work, as you do in the Conclusion section.
We understand this comment. Our decision to keep a Results and Discussion as a single section responds to the brief discussions following the CPR Design Principles and the associated Interviewee quotations. Also, to the content of the narrative in the Conclusions section.
- In the Conclusion section I would like to be highlighted in details role, contribution and possible implications of this research.
Lines 707 to 712 and 721 to 726 have been added. Also, we consider that lines 696-706 reflect the contributions and possible implications of the present work.
- Acronyms: this paper reports a number of acronyms, often and repeatedly cited into the text, strongly affecting its readability. I suggest to add a table with all acronyms and a diagram that shows links between institutions and laws.
Below is the list of acronyms used. We are more than happy to create a Table if the Journal has a specific format requirement for that.
- Common Pool Resource (CPR)
- Nebraska Department of Natural Resources (NeDNR)
- Natural Resource Districts (NRDs)
- Integrated Management Plan (IMP)
- Ground water Management Protection Act (GWMPA)
- Upper Platte Basin (UPB)
- Central Nebraska Public Power and Irrigation District (CNPPID)
- Global Land Data Assimilation System (GLDAS)
- MODerate resolution Imaging Spectroradiometer (MODIS)
- Cooperative Hydrology Study (COHYST)
Specific comments:
- Figure 1: could you use more evident colors?
We modified the colors and increased the contrast to highlight the difference between the High Plains Aquifer and the Platte River Basin. Also, we extended the description of the figure, expanding the description of the FIGURE in the caption. Now, the caption includes the meaning of the map at the bottom of Figure 1. The added text describes the discretization of the state in 23 Natural Resources Districts (NRDs). The NRDs administer the ground water distribution and coordinate with the Nebraska Department of Natural Resources to manage the ground and surface water in the state conjunctively.
- Figure 2: please use different colors, and specify on the map the geographical position of the basin.
The comment is welcome. The colors have been changed to contrast the portion of the basin that is Fully appropriated and over appropriated. The light lines represent the tributaries' boundaries and the state's boundaries' dark and thick lines. This figure complements Figure 1 described in the caption and the comment above.
- Figure 3: it is not clear where the contribution of "Integrated hydrological processes" emerges within the figure. It seems that it should be with "normalizd recharge", that have no definition or links in the text
The comment is correct. We clarified the definition of Normalized Recharge in the methodology (line 287 to Line 299)
- Lines 132-135: could you provide a reference for this statement?
The lines referred to in this comment were moved to the end of the Results and Discussion section (also suggested in the second Broad Comment). The corresponding information is discussed between Lines 623 and 642. We added to this paragraph the experiences of Amaranto et al. (2018, 2019, 2020). They implemented a machine learning technique based on simplifying the regional hydrologic system using evapotranspiration, streamflow, ground water level changes, and rainfall as inputs.
- Lines 139-141: I think that it should be moved in the Conclusion section
The suggestion is welcome and now has been paraphrased and added to lines 707 to 712.
THANK YOU!

Reviewer 3 Report
The work is interesting and covers Platte River Basin (PRB) and Nebraska's Natural Resources. However, the results of the work can be extended to other parts of the world. Despite this, the work has many problems that I have highlighted in the text of the paper

Author Response
Dear Reviewers and Editors,
We appreciate the valuable comments you made by. The sections were reviewed and reorganized based on their comments, leading to a more robust and fluent narrative. All the changes made were tracked and observed in the version “REED et al Hydrology CPR_HYDROLOGY-REPLAY-REV-TRACK.” Below, you can see our answers to each of your comments.
Best regards
Francisco Muñoz-Arriola, PhD
REVIEWER 3
Comments and Suggestions for Authors
Provide a graphical abstract for better understanding.
We submitted a graphical abstract before. We assume it is available in the Editor’s platform for reviewers.
Result section required some figures and tables, please provide your result as a schematic form in some figures.
As part of the manuscript's re-arrangement, now Figure 3 and Table 2 are part of the Results and Discussion section (staring in Line 330).
Materials and Methods should be extended, please use references, formula and also more details about methods and data.
The Methodology was extended. Also, the description of the data has additional details. (see Lines 254 to 310)
Use some newly literature review.
We added references that provide updated views of the topic proposed topic in this manuscript. The content of those references was synthesized. We integrated into the narrative new perspectives in support of the proposed hypothesis, testing, and discussion. (see at the end of the cited references)
This manuscript is a bit far from standard form, and at least it needs 2 (or 3) round revisions.
We cited few references that Reviewer 1 can consult. The manuscript is aligned with other articles in the field. Also, we consider the document responds to the call for papers in this special issue. With all respect, we welcome the valuable comments of the reviewer, but we also value a review that does not require multiple revisions.
Then for now please handle mention above comments. More details in next round will be informed (according to your revised file).
THANK YOU!

Round 2
Reviewer 1 Report
This manuscript is acceptable.